# *WT1* Gene Pathogenic Variants: Clinical Challenges and Treatment Strategies in Pediatric Nephrology—One Center Practice

**DOI:** 10.3390/ijms26083642

**Published:** 2025-04-11

**Authors:** Artur Janek, Andrzej Badeński, Marta Badeńska, Martyna Szuster, Karolina Szymańska-Kurek, Elżbieta Trembecka-Dubel, Maria Szczepańska

**Affiliations:** 1Department of Pediatrics, Faculty of Medical Sciences in Zabrze, Medical University of Silesia in Katowice, ul. 3 Maja 13/15, 41-800 Zabrze, Poland; a.janek97@gmail.com (A.J.); marta.badenska2@gmail.com (M.B.); martynagorka@gmail.com (M.S.); etdubel@interia.pl (E.T.-D.); mszczepanska@sum.edu.pl (M.S.); 2Department of Pediatric Nephrology with Dialysis Division for Children, Independent Public Clinical Hospital No. 1, ul. 3 Maja 13/15, 41-800 Zabrze, Poland; kazanowska14@interia.pl

**Keywords:** Wilms’ tumor suppressor gene 1 (*WT1*), steroid-resistant nephrotic syndrome (SRNS), Wilms’ tumor (WT)

## Abstract

Pathogenic variants in the Wilms’ tumor suppressor gene 1 (*WT1* gene) can lead to serious disorders within the kidney and urogenital system, including chronic kidney disease. There is still much uncertainty regarding the optimal management of diseases caused by *WT1* dysfunction, posing a challenge for physicians caring for these patients. The aim of our study is to present experiences related to the course and treatment of patients with confirmed *WT1* pathogenic variants. Data from seven patients (five girls, two boys), who were at the age of 4.8 ± 5.1 years (0.3–14 years) at their first admission and were treated between 1997–2022, were analyzed. The analysis included each patient’s age at the day of diagnosis, anthropometric measurements, comorbidities, and laboratory and genetic test results, as well as their treatment, oncological procedures, and performed surgeries. Wilms’ tumor was the first manifestation of the disease in three patients. Arterial hypertension was diagnosed in three patients, and anemia in four. Treatment of patients with nephrotic syndrome included glucocorticosteroid therapy (GCS), calcineurin inhibitors (CNIs), and mycophenolate mofetil (MMF). Nephrectomy was performed in five children, while kidney transplantation was carried out in two patients. An interdisciplinary approach to *WT1* gene pathogenic variants, including early diagnosis, individualization, regular monitoring of treatment, and oncological vigilance, is crucial for improving prognosis and ensuring proper care for patients with nephrological manifestations of *WT1* gene region disorders. Furthermore, for a comprehensive understanding of the scope of this disease and the development of effective therapy methods, continued research on the clinical manifestations of *WT1* pathogenic variants is essential.

## 1. Introduction

One of the causes leading to a range of urogenital disorders is pathogenic variants in the Wilms’ tumor gene 1 (*WT1*). The *WT1* gene, also known as the Wilms’ tumor suppressor gene, located on chromosome 11, spans 50 kb of genomic DNA, consists of 10 exons, and generates 3 kb mRNA which, along with four Krüppel-type zinc fingers, forms a transcription factor that plays a significant role in the development of the urinary system and kidneys [1]. The main functions of the *WT1* gene in relation to the kidneys include the following: self-renewal and proliferation of metanephric mesenchyme (MM), regulation of MM’s differentiation into epithelial cells, and activation of podocyte-specific transcription factors defining epithelial cells’ identity and function [1].

A wide spectrum of renal symptoms, including oncogenic characteristics, may result from *WT1* expression dysfunction, ultimately leading to chronic kidney disease (CKD) [1,2,3]. Specific pathogenic variants in the *WT1* gene result in the following variable phenotypes: Frasier syndrome (FS), Denys-Drash syndrome (DDS), steroid-resistant nephrotic syndrome (SRNS), and disorders in genital development [4,5]. FS, caused by a pathogenic variant in intron 9 of the *WT1* gene, is characterized by progressive glomerulopathy, gonadoblastoma developing in the second decade of life, and complete gonadal dysgenesis (CGD) in 46,XY individuals [4,6]. DDS, linked to pathogenic variants in exons in the zinc finger region, is associated with rapidly progressive kidney failure, a high risk of Wilms’ tumor, and congenital anomalies of the genitourinary track and/or 46,XY disorder of sex development (DSD) [3,7,8]. According to the recommendations of the IPNA, reported by Trautmann et al., SRNS is defined as a nephrotic syndrome for which remission is not achieved with steroid therapy within 4–6 weeks [9,10,11]. Due to the steroid resistance of nephrotic syndrome in *WT1* disorder, the standard first-line treatment for SRNS involves the gradual withdrawal of steroid therapy and the initiation of immunosuppressive therapy, along with nephroprotective treatment using ACE inhibitors and Angiotensin Receptor Blockers (ARBs) [12]. However, in cases where a genetic basis for SRNS is confirmed, CNI therapy is not recommended [13]. Proteinuria occurring in genetically determined nephrotic syndrome is a significant predisposing factor for end-stage kidney failure (ESRD), for which kidney transplantation is the treatment of choice [10]. There are still many uncertainties regarding the proper management of diseases resulting from pathogenic variants in the *WT1* gene. This poses a challenge for physicians in terms of monitoring, treatment, and preventing progression. Due to the development of molecular medicine and the performance of research enabling the understanding of genomic mechanisms, there is a great chance of discovering targeted and specific treatment for specific genetically based diseases [14].

### Aim

In our review, we will share our experience regarding the course of the disease and treatment of patients with confirmed *WT1* pathogenic variants.

## 2. Results

Laboratory tests conducted during their first hospitalization in the Department of Pediatric Nephrology with Dialysis Division showed increased levels of urea and cholesterol in all children included in the study. Additionally, decreased serum albumin levels and borderline total protein values were observed in all patients. Six patients exhibited proteinuria of varying magnitude, while the criteria for nephrotic syndrome were met in four patients (57.1%). SRNS was diagnosed in three (42.9%) of the patients included in the study. In one patient (patient 1), due to anuria and kidney insufficiency in their first days of life, urine biochemical tests were not conducted. Furthermore, anemia of varying severity was found in four patients (57.1%) and arterial hypertension was also diagnosed in four patients (57.1%). Wilms’ tumor was detected in four patients (57.1%) during the course of the disease. Consequently, they underwent oncological treatment, specifically chemotherapy, following the SIOP 2001 protocol. Complete gonadal dysgenesis (CGD) was observed in one 46,XY individual presenting the female phenotype. The remaining patients included in the study did not present any abnormalities concerning their genital organs.

With regard to the treatment, steroid therapy was initiated in five patients (71.4%). Treatment with CNIs (cyclosporine, tacrolimus) was administered to three patients (42.9%), and MMF therapy after kidney transplantation was used for two patients (28.6%). Due to the progression of oncological disease, nephrectomy was performed in five children (71.4%). Moreover, kidney transplantation was conducted in two patients (28.6%) within 1.5 and 2 years after the completion of oncological treatment.

Genetic diagnostics were performed in collaboration with the Clinical Genetics Unit of the Department of Biology and Medical Genetics at the Medical University of Gdańsk. The individual genetic variants in the *WT1* gene are listed in Table 1.

Genetic testing was performed on sixty-four children treated for SRNS and CNS in the center between 1997–2022, including screening for pathogenic variants and polymorphisms in the *WT1* gene. A pathogenic *WT1* variant was identified in seven patients. In four additional children who were not tested for *WT1* variants, the clinical picture clearly indicated an alternative genetic etiology. These included a patient with a pathogenic variant in *LMX1B* and a positive multi-generational family history; a patient diagnosed with Charcot-Marie-Tooth syndrome; patient 3, who had a pathogenic *SMARCAL1* variant; and patient 4, who had a positive family history of atypical hemolytic uremic syndrome (aHUS). In this patient, thrombotic microangiopathy (TMA) was initially ruled out, followed by testing for a pathogenic variant in the *C3* gene, which was negative. Due to the presence of isolated proteinuria, a diagnosis of congenital nephrotic syndrome (CNS) was performed.

The analysis of the correlation between the type of *WT1* gene variant and the clinical course revealed several associations. Patients carrying the c.1399C>T variant (patients 2 and 3) exhibited a severe disease course, including steroid-resistant nephrotic syndrome (SRNS), progression to end-stage renal disease (ESRD), and the necessity for nephrectomy. Both patients were diagnosed with Wilms’ tumor, and patient 3 underwent kidney transplantation. The c.1447+5G>A splice site variant, identified in patients 5 and 7, was associated with variable clinical manifestations. Despite its classification as pathogenic, the patients differed in disease expression. Patient 5 presented with isolated proteinuria and responded solely to nephroprotective therapy. In contrast, patient 7 developed SRNS, ESRD, and anemia, indicating a more severe clinical course despite carrying the same genetic variant. Patient 6, harboring the novel missense variant c.1337C>T, experienced severe disease progression, including SRNS, ESRD, bilateral nephrectomy, and complete gonadal dysgenesis (CGD). The phenotype in this case suggests a potentially pathogenic character of this variant, which is currently classified as a variant of uncertain significance (VUS) because of insufficient data and a lack of stronger evidence.

## 3. Discussion

The diagnosis and treatment of nephrological disorders in children with *WT1* gene pathogenic variants remains one of the most complex challenges for clinicians. This condition is associated with a diverse spectrum of diseases, ranging from steroid-resistant nephrotic syndrome to gonadal disorders and oncological processes [4,5,15]. The expression of the *WT1* gene results in the induction of podocalyxin, a protein responsible for the regulation of nephron formation during embryonic life and the maintenance of glomerular filtration function [16,17]. Disorders related to the *WT1* pathogenic variants are lifelong, and glomerular damage along with an impaired filtration barrier can lead to nephropathy [17]. Pathogenic variants in the *WT1* gene lead to glomerulopathies, with a histopathological picture including focal and segmental glomerulosclerosis (FSGS) and diffuse mesangial sclerosis [18]. According to Lipska et al., FSGS primarily affects older children, while diffuse mesangial sclerosis is predominantly observed in children under the age of two. To date, approximately 500 cases of nephrological manifestation during the course of *WT1* gene pathogenic variants have been described. No specific geographical distribution or founder variant has been identified in a particular population [19]. The lack of effective preventive strategies and the difficulties associated with the diagnosis of *WT1* pathogenic variants constitute a significant limitation in the therapy of correlated diseases.

*WT1* disorders are inherited in an autosomal dominant manner [19]. Most confirmed pathogenic variants in individuals with clinical manifestations appear to occur de novo. However, the literature reports inheritance from heterozygous parents carrying the pathogenic *WT1* variant, leading to the onset of the disease in their offspring [19,20,21,22,23]. The current data on penetration are limited due to the absence of routine testing for pathogenic *WT1* variants in asymptomatic parents of affected children. If no pathogenic variant is detected in both parents, the risk of inheritance in siblings is slightly greater compared to the general population due to the possibility of gonadal mosaicism. In their study, Kaneko et al. noted the dependence of *WT1* gene penetration on the sex of the heterozygous parent. It was found that there is greater penetration of the pathogenic *WT1* variant originating from the father compared to the variant inherited through the maternal line [22]. Early recognition of genetic forms of kidney disorders appears to be a crucial strategy to avoid or shorten unnecessary and potentially harmful treatment and to start the appropriate therapeutic process sooner.

The molecular characteristics of *WT1* disorders determine the clinical type and renal survival time, indicating a high genotype–phenotype correlation [17]. One of the clinical manifestations of pathogenic variants in the *WT1* gene is sex reversal, as presented by one of the patients treated in our center. A phenotypical girl with an XY karyotype was genetically diagnosed during her first year of life. However, Arroyo-Parejo Drayer et al. point out that phenotypic females with complete 46XY sex reversal and progressive chronic kidney disease are often diagnosed too late—the diagnosis of Frasier syndrome is only made when the diagnostic evaluation of delayed sexual maturation is extended to include karyotyping [24]. Such reports indicate the necessity of karyotyping in every patient diagnosed with a pathogenic variant in the *WT1* gene. Currently, the greatest limitation of routine genetic testing is still its cost and availability. A more cost-effective and rapid alternative to conventional karyotyping is quantitative-fluorescent polymerase chain reaction (QF-PCR), a technique used in perinatal diagnostics that can also determine gender [25]. According to Stark et al., in searching for genetic causes of kidney failure or nephrotic syndromes, genetic sequencing may be cheaper and more beneficial for the patient than cascading laboratory and imaging tests, preceding genetic tests [26,27].

In the context of kidney diseases associated with *WT1* gene pathogenic variants in children, it is essential to understand that this disorder is most often characterized by congenital nephrotic syndrome, SRNS with an onset in childhood, and progressive glomerulopathy that is unresponsive to standard treatment [28]. Although nephrological manifestations of *WT1* disorders are considered exceptionally rare, the broad range of symptoms prompts discussion about the existence of many unrecognized cases. Additionally, the occurrence of Wilms’ tumor, as reported by Lipska et al. and Lehnhardt et al., may precede SRNS by up to 4 years, but as the first observed manifestation of a pathogenic variant in the *WT1* gene, it may precede the development of SRNS by up to 10 years after the completion of oncological treatment [3,21]. Among the patients we described, in three out of four children diagnosed with Wilms’ tumor, it was indeed the Wilms’ tumor that was the first manifestation of pathogenic variants in the *WT1* gene. Only in one of the patients under observation was Wilms’ tumor diagnosed 2 years after the recognition of the *WT1* pathogenic variant.

In our study, steroid-resistant nephrotic syndrome was identified in two girls and one boy under observation. There are reports suggesting a stronger association between SRNS occurrence in girls and *WT1* pathogenic variants compared to boys, as indicated in the study by Li et al., where among 16 patients with SRNS, 15 were female. It is worth noting that all patients underwent karyotype analysis during the diagnostic process [29]. The rate of genetic detection and the predominant genes associated with CNS differ across various countries and regions around the world. Despite the lack of precise statistics among the European population, Trautmann et al. identified the pathogenic *WT1* variant as the third most common cause of SRNS [11]. According to Eujin Park, *WT1* defect was the most frequently detected pathogenic variant in a study of congenital nephrotic syndrome (CNS) [30], where a total of 57 genes were analyzed, while Zhu et al., in a large cohort study of 283 patients from the Chinese population, identified *WT1* gene pathogenic variants as the most common cause of SRNS [31], with 26 genes being examined. According to Li et al., SRNS inevitably leads to ESRD within 1–11 years of diagnosis [29], whereas Boyer et al. reported that the progression from genetically conditioned CNS to ESRD occurs much faster, within weeks to months [12,19]. Among the symptoms of kidney disorders associated with *WT1* gene pathogenic variants, the most common and widespread symptom is proteinuria occurring from birth [19,24]. In our center, almost 100% of the children studied exhibited proteinuria of varying severity, while in the study by Arroyo-Parejo Drayer et al., proteinuria was present in 92% of the subjects [24]. Furthermore, in our observation, anemia was detected in four patients. Chen et al. also demonstrated correlations between SNRS, co-existing tumors and malformations of the urinary system, and mild anemia, which appears to be a consequence of impaired erythropoietin (EPO) production by damaged kidneys [17].

The preferred approach during the initial episode of nephrotic proteinuria is steroid therapy [32,33]. Trautmann et al. suggest tapering off steroid therapy after establishing a diagnosis of SRNS and discontinuing steroid therapy after 6 months, while recommending the introduction of calcineurin inhibitors as the primary treatment. However, calcineurin inhibitors show reduced efficacy in genetically determined SRNS, necessitating a personalized approach based on the specific genetic profile [11]. In cases where there is no response to immunosuppressive therapy with CNIs in the treatment of monogenic SRNS, discontinuation of CNI treatment is suggested due to potential decreased benefit relative to harm [11]. Yue et al. found that the use of CNIs was effective in patients with pathogenic variants involving missense and nonsense variants in the *WT1* gene [13]. According to Faul et al., cyclosporine (CsA), like CNIs, blocks the calcineurin-dependent dephosphorylation of synaptopodin, which is involved in podocyte damage [31,34]. However, Bensman et al. found that CsA therapy does not slow the progression of the disease to ESRD [35]. In the study by Büscher et al. examining the impact of CsA therapy on the progression of ESRD during the course of CNS/SRNS, no significant difference in disease progression with or without CsA therapy was shown [36]. For our patients, CNI treatment was unsuccessful.

Given the rapid progression of disorders related to *WT1* gene pathogenic variants, including SRNS’s progression to ESRD, kidney transplantation seems to be the most effective treatment to improve the clinical condition of the patient [17,24,37,38]. According to Chen et al., the best treatment for hereditary nephropathy is kidney transplantation, and the recurrence of renal disease post-transplant due to hereditary nephropathy is very rare, as observed over a 2.1-year follow-up of four patients who received kidney transplants for nephropathy associated with *WT1* gene pathogenic variants [17]. In the study by Roca et al., in five children diagnosed with *WT1* pathogenic variants and established DDS, no recurrence of nephropathy associated with the *WT1* gene pathogenic variants was observed after kidney transplantation over an average observation period of 16 years [38]. In the case review by Arroyo-Parejo Draye, it was noted that kidney transplantation provided long-term patient survival [24]. Due to the high risk of developing Wilms’ tumor, performing a nephrectomy is also considered. This issue was significantly highlighted by Arroyo-Parejo Drayer et al., who noted that earlier historical consent for prophylactic nephrectomy in children diagnosed with *WT1* gene pathogenic variants was indicated [24]. Today, however, there are no specific recommendations for children with *WT1* pathogenic variants in favor of bilateral prophylactic nephrectomy, and more recently, kidney-sparing surgery is recommended in cases of unilateral Wilms’ tumor [24]. Among the patients included in the study at our center, kidney transplantation was performed on two patients (Table 2) in 2017 (patient 1) and 2007 (patient 3), resulting in a noticeable improvement in their clinical condition.

Potential new directions for research into therapeutic strategies for *WT1* disorders were presented in the work by Imeri et al. In a mouse study, the impact of the loss of sphingosine kinase 2 (Sphk2) activity on podocyte function was evaluated. It was found that inhibition of sphingosine kinase 2, which regulates sphingosine-1-phosphate, a biologically active sphingolipid regulating various cellular functions in the kidneys, leads to increased expression of the *WT1* gene and nephrin in podocytes. Increased nephrin expression affects the stabilization of the glomerular membrane, reducing albuminuria and protecting against podocytopathy [39]. New therapeutic targets, such as the aforementioned selective Sphk2 inhibitors potentially affecting *WT1* gene expression and reducing proteinuria [17,39], as well as innovative strategies like stem cell therapy [40], highlight promising directions for future research into the treatment of nephropathy associated with *WT1* pathogenic variants.

## 4. Materials and Methods

Data from 7 patients treated between 1997 and 2022 in the Department of Pediatric Nephrology with Dialysis Division for Children at the Medical University of Silesia Hospital No. 1 in Zabrze were analyzed. The average observation period was 9.5 ± 5.6 years. Among the children included in the study, there were 5 girls (71.4%) and 2 boys (28.6%). Patient 6, despite having an XY karyotype, presented with a female phenotype and self-identified as female. Therefore, in Table 1, Table 2,Table 3 and Table 4, the patient has been classified as female. The average age of the patients at the time of their first hospitalization in the Department of Pediatric Nephrology was 4.8 ± 5.1 (0.3–14) years. During the analysis of medical records, age at disease diagnosis, anthropometric measurements (Table 3), laboratory tests (Table 4), observation time, treatment administered, oncological and surgical procedures performed, and accompanying diseases, genetic tests were taken into consideration.

The diagnosis of a *WT1*-related disorder was established in the patients based on the presence of a pathogenic or likely pathogenic variant with clinically consistent data in the *WT1* gene, identified through molecular genetic testing. The classification criteria for specific variants according to the American College of Medical Genetics and Genomics (ACMG) [41] are presented in Table 4.

Between 1997 and 2022, a total of 68 children were treated at the center for steroid-resistant nephrotic syndrome (SRNS) and congenital nephrotic syndrome (CNS). Due to the limited availability of genetic testing prior to 2020, a 21-gene panel was used (*NPHS1*, *NPHS2*, *PLCE1 [NPHS3]*, *LAMB2*, *SMARCAL1*, *ADCK4*, *COQ2*, *COQ6*, *PDSS2*, *MYO1E*, *PTPRO*, *CD2AP*, *GMS1*, *COL4A3*, *COL4A4*, *WT1*, *LMX1B*, *INF2*, *TRPC6*, *ACTN4*, *COL4A5*). Since 2020, a broader panel of 50 genes has been implemented (*ACTN4*; *ADCK4*; *ANLN*; *APOL1*; *ARHGAP24*; *ARHGDIA*; *AVIL*; *CD2AP*; *COL4A3*; *COL4A4*; *COL4A5*; *COQ2*; *CRB2*; *DGKE*; *DLC1*; *EMP2*; *FAN1*; *FAT1*; *FN1*; *INF2*; *ITGA3*; *KANK1*; *KANK2*; *KANK4*; *LAMB2*; *LMX1B*; *MAFB*; *MAGI2*; *MYH9*; *MYO1E*; *NPHS1*; *NPHS2*; *NUP107*; *NUP133*; *NUP160*; *NUP205*; *NUP85*; *NUP93*; *OSGEP*; *PLCE1*; *PTPRO*; *SCARB2*; *SGPL1*; *SMARCAL1*; *TBC1D8B*; *TRPC6*; *TTC21B*; *WDR73*; *WT1*; *XPO5*).

## 5. Conclusions

In summary, pathogenic variants in the *WT1* gene, associated with multiple organ manifestations, require an interdisciplinary approach that includes early diagnosis together with genetic testing, individualization of treatment based on the clinical profile of the patient, and monitoring and early detection of tumors such as Wilms’ tumor and gonadoblastoma. Novel pathogenic variants in the *WT1* gene continue to be discovered, contributing to the heterogeneous clinical course observed in affected patients. However, predicting disease progression remains challenging, as the same *WT1* pathogenic variant may present with variable phenotypic expression across different individuals. Early recognition and a personalized approach to treatment are crucial for improving the prognosis of patients with *WT1* disorders. Further research is also necessary to understand the full spectrum of this disease and to develop more effective treatment methods.

## Figures and Tables

**Table 1 ijms-26-03642-t001:** The genetic diagnosis of the patients included in the study.

Patient		Gene	Protein Change	Variation	Variant Type	Classification	ACMG Classification Criteria **
Male	1	*WT1*	-	c.1339+1G>A	Splice variant	Likely pathogenic	PVS1, PM2
	2	*WT1*	Arg467Trp	c.1399C>T (before c.1384C>T)	Missense variant	Pathogenic	PS1, PM2, PP3
Female	3	*WT1*	Arg467Trp	c.1399C>T (before c.1384C>T)	Missense variant	Pathogenic	PS1, PM2, PP3
	4	*WT1*	Arg458X	c.1372C>T	Nonsense variant	Likely pathogenic	PVS1, PM2
	5	*WT1*	Not known	c.1447+5G>A (before 1432+5G>A)	Splice variant	Pathogenic	PVS1, PP3, PP4
	6	*WT1*	Thr446Ile	c.1337C>T	Missense variant	VUS *	PM2, PP3
	7	*WT1*	-	c.1447+5G>A (before 1432+5G>A)	Splice variant	Pathogenic	PVS1, PP3, PP4

* Variant of uncertain significance. ** American College of Medical Genetics and Genomics (ACMG) classification criteria: PVS1—Very Strong Evidence of Pathogenicity for loss-of-function (LoF) variants (e.g., nonsense, canonical ±1 or ±2 splice sites, frameshift); PS1—Strong: the same amino acid change as a previously established pathogenic variant; PM2—Moderate: absent from controls in population databases (e.g., gnomAD); PP3—Supporting: multiple lines of computational evidence support a deleterious effect (e.g., SIFT, PolyPhen, MutationTaster); PP4—Supporting: patient’s phenotype or family history is highly specific to a disease with a single genetic etiology.

**Table 2 ijms-26-03642-t002:** The clinical course parameters assessed in patients with *WT1* gene pathogenic variants.

Patient		Year of Diagnosis *	Observation Period [Years]	Cancer History	Renal Dysfunction	Applied Treatment of NS	Nephrectomy	Additional Disorders	Kidney Transplantation
Male	1	2014	4	Wilms’ tumor	ESRD	GCS + MMF + CNIs	Bilateral nephrectomy	Cryptorchidism	Yes
	2	2008	14	Wilms’ tumor	SRNS; ESRD	GCS	Right-sided nephrectomy	Hypospadias; cryptorchidism	No
Female	3	2003	18	Wilms’ tumor	SRNS; ESRD	GCS + MMF + CNIs	Bilateral nephrectomy	Anemia at diagnosis	Yes
	4	2017	5	Wilms’ tumor	Proteinuria	GCS + nephroprotection	Right-sided nephrectomy + left-sided heminephrectomy	-	No
	5	2013	8	No	Proteinuria	Nephroprotection only **	No	-	No
	6 ****	2012	2	No	SRNS; ESRD	GCS	Bilateral nephrectomy	Anemia at diagnosis; bilateral vesicoureteral reflux; CGD; FS	Outcome shortage ***
	7	1997	15	No	SRNS; ESRD	GCS + CNIs	No	Anemia	Outcome shortage ***

ESRD—end-stage kidney failure; SRNS—steroid-resistant nephrotic syndrome; CGD—complete gonadal dysgenesis; FS—Frasier syndrome; GCS—glucocorticosteroids; MMF—mycophenolate mofetil; CNIs—calcineurin inhibitors. * The year of diagnosis is considered the first manifestation of the disease due to the *WT1 gene* pathogenic variants. In the cases of patients 1, 2, 3, and 4, this was the diagnosis of Wilms’ tumor, whereas in patients 5, 6, and 7, it was proteinuria. ** Nephroprotection included treatment with ACE inhibitors (ACEI). *** The patient passed away before transplantation. **** A phenotypic female presenting with a 46,XY karyotype.

**Table 3 ijms-26-03642-t003:** The anthropometric parameters of patients with *WT1* gene pathogenic variants during their first hospitalization in the department.

Patient		Gender Genotype	Age * [Years]	Height [cm]	Height Percentile	Weight [kg]	Weight Percentile	BMI [kg/m^2^]	Hypertension
Male	1	XY	0.5	77	25–50	10.5	50–75	17.6	Yes
	2	XY	8	117	<3	19.8	<3	14.5	No
Female	3	XX	14	156	10–25	42	10	17.3	No
	4	XX	0.3	71	>97	8	>97	15.9	Yes
	5	XX	2.9	86	<3	11.2	<3	15.1	Yes
	6 **	XY	0.3	57	<3	5.5	<3	16.9	No
	7	XX	7	126	50–75	27.3	75	17.2	Yes

* Age of the patient at the time of their first hospitalization in the Nephrology Department. ** Complete gonadal dysgenesis (CGD).

**Table 4 ijms-26-03642-t004:** A comparison of selected biochemical blood parameters’ levels during the course of *WT1*-related disorders in the studied patients during their first hospitalization in the department.

Patient		Creatinine [µmol/L]	Urea [mmol/L]	Total Cholesterol [mmol/L]	Total Protein [g/L]	Serum Albumin [g/L]
Male	1	399	12.2	5.5	52.6	36.04
	2	66.4	9.15	4.74	67.7	37.7
Female	3	557	31.9	6.01	57.6	33.32
	4	40	1.2	5.1	57.6	37.23
	5	30	3.1	4.88	63.5	40.98
	6	417	10.7	4.38	53.2	39.1
	7	53	4.87	6.18	56.9	0.51

**Reference values:** Creatinine [µmol/L] 1–12 months 6–23, 1–6 years 6–57, 7–12 years 6–74, 13–18 years 6–100; urea [mmol/L] <3 years 1.8–6, 3–14 years 2.5–6; total cholesterol [mmol/L] <9.4; total protein [g/L] 6–12 months 44–76, 1–3 years 56–75, 3–18 years 60–80; serum albumin [g/L] 35–50.

## Data Availability

Data are contained within the article.

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
