# Peer review of "WT1 Gene Pathogenic Variants: Clinical Challenges and Treatment Strategies in Pediatric Nephrology—One Center Practice"

_ijms, 2025, doi:10.3390/ijms26083642_

Round 1

Reviewer 1 Report

Comments and Suggestions for Authors

Dear Editor,

Thank you for the opportunity to review the manuscript entitled “ WT1 gene pathogenic variants: clinical challenges and treatment strategies in pediatric nephrology – one center practice”, by Artur Janek at all.

The clinical significance of pathogenic variants of WT1 gene expression is a rare but important issue. As it was mentioned by authors, approximately 500 cases have been described to date. The clinical consequences of these abnormalities include genetic syndromes, glomerulopathies, end-stage renal disease, and Wilms tumor.

I agree with the Janek at all., that there are currently no established guidelines for the management of patients with specific WT1 gene variants. Therefore, the analysis of seven cases presented in the manuscript has significant clinical value worth of publication.

However, after lecture of the manuscript, I would like to raise one concern regarding the presentation of the results. This section is too brief and without potential description of potential correlations between specific WT1 gene variants and clinical manifestations, response to treatment, and patient outcomes. I recommend expanding this part of the manuscript.

Apart from this comment, I have no further remarks. I believe that after revision of the results section, the article will be suitable for publication in the International Journal of Molecular Sciences.

Author Response

Reviewer 1

We appreciate the Reviewer’s positive assessment of the clinical relevance of the cases presented. In response to the suggestion, we have substantially expanded the Results section to provide a more comprehensive description of potential correlations between specific WT1 gene variants, clinical manifestations, treatment responses, and patient outcomes.

Reviewer 2 Report

Comments and Suggestions for Authors

Dear Authors, 

It was a pleasure to review this presentation of a small number of cases with WT1 genetic variants.  It  is always interesting to get the perspective of a genetic condition in different populations and from this point of view I think your article is even more interesting.  I appreciated the  clear abstract and the detailed Discussion  section.  However, I had a few suggestions to improve this presentation: 

  1. The Materials and Methods section does not include details about the reference sequence used to interpret the results of the genetic test or about the classification criteria for these genetic variants.  WT1 reference sequence has changed a lot during the time considered in the article and some variants may have changed their names in the meantime.
  2. Table 4 would benefit from a few changes:
    1. The first variant: 2c.1339+1G>A is not accurately transcribed. It would be more accurate to write it as 'c.1339+1G>A'.   
    2. The same variant type is not 'unknown'  but it is a splice variant, changing the conserved donor site.  Moreover, this was previously reported in an article cited here (reference 4) and it is located in a relevant part of the gene, meaning that this is more likely to be re-classified. The nomenclature used for this variant also needs to be updated, based on the most recent reference sequence. 
    3.  The other variants in this list, also need updated nomenclature and classification.
    4. The variant corresponding to patient 4 (written as 'R458') needs further clarification regarding the amino acid change.
    5. Usually, those variants classified as 'variants of unknown significance' or VUSs  are not considered valid to guide clinical management.   However, it is understandable  that in some cases, the phenotype is strongly suggestive for the gene, even if we required level of evidence is not achieved.  If this is the case, this will need to be mentioned in the table, to include the reason why this variant was considered as relevant for the patient.
  3.   It would be helpful to add information regarding classification criteria for each variant, either as part of Table 4, or separately.
  4.   Table 3:   I think describing patients 6 as female is not entirely accurate and I think this patient should be mentioned separately.
  5. Discussion: row 43:   I would argue that not only karyotyping but also QFPCR are useful techniques for gender determination.  This  is because QFPCR can be done quicker and cheaper than the karyotype.

 I hope you will find this observations helpful for further improving your article.

Author Response

Reviewer 2

Thank you for your thoughtful review and suggestions. The following changes have been made:

  1. Materials and Methods section:

We have included detailed information on the reference sequences employed for WT1 variant interpretation, along with the classification criteria utilized. Additionally, historical changes in variant nomenclature have been acknowledged and appropriately clarified.

  1. Table 4:

   - The transcription of the variant c.1339+1G>A has been corrected to reflect the accurate nomenclature.

   - All variants have been re-evaluated and updated according to the latest reference sequence and nomenclature standards.

   - Splice-site variants have been appropriately reclassified and their potential impact discussed in Results section.

   - For variants of uncertain significance (VUS), additional justification has been provided both in the table and the Results section, outlining their clinical relevance based on phenotypic correlation.

  1. Variant classification criteria:

Classification criteria by American College of Medical Genetics and Genomics (ACMG) for each variant have been added directly to Table 4.

  1. Table 3 – Patient 6:

With full respect to the reviewer’s remark, we have decided to maintain the designation of patient 6 as "female" in Tables 1–4, due to the patient's consistent female gender identity and phenotypic presentation. To ensure clarity and transparency, an explanatory footnote has been added below each table, and the rationale is discussed in the Materials and Methods section. The patient’s classification reflects both medical and ethical considerations related to gender identity.

  1. Discussion section:

The value of QF-PCR in gender determination has been acknowledged and included in the discussion.

Reviewer 3 Report

Comments and Suggestions for Authors

The authors analyzed data from 7 patients treated in the Department of Pediatric Nephrology with the Dialysis Division for Children at Hospital No. 1 of the Medical University of Silesia in Zabrze between 1997 and 2022. The selective presentation of 7 patients with their clinical course does not provide new information beyond what has been widely known for a long time in literature.

The authors comparatively discuss various aspects of disease progression and treatment in a group of children with WT1 gene alterations in relation to children without WT1 gene alterations. Unfortunately, this does not result from the study's findings.

  1. Therefore the authors are requested to provide the total number of children treated for SRNS and CNS in the center between 1997-2022.
  2. Additionally, please specify the number of children in whom WT1 mutations and polymorphisms were examined and how many had a positive result. This will allow for obtaining data on the frequency of WT1 polymorphisms in the group of children with SRNS and CNS.

Despite the lack of precise statistics in the European population, Trautmann et al. identify the pathogenic WT1 variant as the third most common cause of SRNS. According to Eujin Park, the WT1 defect was the most frequently detected pathogenic variant in congenital nephrotic syndrome (CNS), where a total of 57 genes were analyzed.

  1. Therefor the authors are requested to supplement the "Materials and Methods" section with details on how WT1 gene alterations were analyzed.
  2. Additionally, the authors should provide information on whether other genes involved in CNS or SRNS were also examined.

Despite the lack of precise statistics in the European population, Trautmann et al. identify the pathogenic WT1 variant as the third most common cause of SRNS. According to Eujin Park, the WT1 defect was the most frequently detected pathogenic variant in congenital nephrotic syndrome (CNS), where a total of 57 genes were analyzed.

  1. Therefor the authors are requested to supplement the "Materials and Methods" section with details on how WT1 gene alterations were analyzed.
  2. Additionally, the authors should provide information on whether other genes involved in CNS or SRNS were also examined.

Out of the 7 observed variants, only two are described as pathogenic. The authors must provide additional data on the impact of the other variants with unknown significance on the function of the WT1 protein.

After supplementing the above data, the discussion and results should be revised accordingly. The conclusions  do not follow from the results of the study and must be revised.

In Table 2, it would be useful to include the reference ranges for each parameter.

In Table 3 and 5, please add "Kidney Transplantation." Below the table, please explain the abbreviations.

Author Response

Reviewer 3

We thank you for your critical assessment and acknowledge the need for contextual epidemiological data. The following revisions have been made:

  1. Patient Population Overview:

We have added information on the total number of children treated for SRNS and CNS in our center between 1997 and 2022.

  1. Genetic Testing:

We have expanded the Materials and Methods section to provide detailed information on the number of patients tested for WT1 gene variants, the number of positive findings, and the observed frequency within the study cohort. Additionally, we have clarified whether other known genes associated with CNS/SRNS were analyzed and explained the rationale behind the chosen genetic testing strategy.

  1. Impact of VUS:

Where applicable, we have added interpretive notes on the possible impact of VUS on WT1 protein function in Methods section.

  1. Discussion and Conclusions:

We have revised the conclusions to ensure they are consistent with and accurately reflect the findings presented in the results section.

  1. Tables:

   - Reference ranges have been added to Table 2.

   - “Kidney Transplantation” has been included in Tables 3.

   - All abbreviations have been explained in footnotes below the respective tables.

Round 2

Reviewer 3 Report

Comments and Suggestions for Authors

The authors have enhanced the paper by addressing the reviewer’s comments. The paper can be published in its current form.